# DeepLungNet: An Effective DL-Based Approach for Lung Disease Classification Using CRIs

**Naeem Ullah [1], Mehrez Marzougui [2], Ijaz Ahmad [3] and Samia Allaoua Chelloug [4],***

[1] Department of Software Engineering, University of Engineering and Technology, Taxila 4400, Pakistan
[2] College of Computer Science, King Khalid University, Abha 61413, Saudi Arabia
[3] Institute of Computer Sciences and Information Technology (ICS/IT), Agriculture University, Peshawar 25130, Pakistan
[4] Department of Information Technology, College of Computer and Information Sciences, Princess Nourah bint Abdulrahman University, Riyadh 11671, Saudi Arabia
* Correspondence: sachelloug@pnu.edu.sa

**Abstract:** Infectious disease-related illness has always posed a concern on a global scale. Each year, pneumonia (viral and bacterial pneumonia), tuberculosis (TB), COVID-19, and lung opacity (LO) cause millions of deaths because they all affect the lungs. Early detection and diagnosis can help create chances for better care in all circumstances. Numerous tests, including molecular tests (RT-PCR), complete blood count (CBC) tests, Monteux tuberculin skin tests (TST), and ultrasounds, are used to detect and classify these diseases. However, these tests take a lot of time, have a 20% mistake rate, and are 80% sensitive. So, with the aid of a doctor, radiographic tests such as computed tomography (CT) and chest radiograph images (CRIs) are used to detect lung disorders. With CRIs or CT-scan images, there is a danger that the features of various lung diseases' diagnoses will overlap. The automation of such a method is necessary to correctly classify diseases using CRIs. The key motivation behind the study was that there is no method for identifying and classifying these (LO, pneumonia, VP, BP, TB, COVID-19) lung diseases. In this paper, the DeepLungNet deep learning (DL) model is proposed, which comprises 20 learnable layers, i.e., 18 convolution (ConV) layers and 2 fully connected (FC) layers. The architecture uses the Leaky ReLU (LReLU) activation function, a fire module, a maximum pooling layer, shortcut connections, a batch normalization (BN) operation, and group convolution layers, making it a novel lung diseases classification framework. This is a useful DL-based method for classifying lung disorders, and we tested the effectiveness of the suggested framework on two datasets with a variety of images from different datasets. We have performed two experiments: a five-class classification (TB, pneumonia, COVID-19, LO, and normal) and a six-class classification (VP, BP, COVID-19, normal, TB, and LO). The suggested framework's average accuracy for classifying lung diseases into TB, pneumonia, COVID-19, LO, and normal using CRIs was an impressive 97.47%. We have verified the performance of our framework on a different publicly accessible database of images from the agriculture sector in order to further assess its performance and validate its generalizability. This study offers an efficient and automated method for classifying lung diseases that aids in the early detection of lung disease. This strategy significantly improves patient survival, possible treatments, and limits the transmission of infectious illnesses throughout society.

**Keywords:** COVID-19; pneumonia; tuberculosis; DeepLunNet deep learning



## 1. Introduction

Each year, thousands of people suffer lung diseases and eventually die due to their illnesses; some of these diseases include lung opacity (LO), pneumonia, COVID-19, tuberculosis (TB), bacterial pneumonia (BP), and viral pneumonia (VP) [1]. Every year, the ratio is anticipated to rise [2]. According to the WHO, the three diseases that kill the most people

worldwide are COVID-19, pneumonia, and TB [3]. There are 450 million afflicted individuals. Additionally, there are more cases involving minors (657 out of 1000). Furthermore, the rapid development of COVID-19 patients has put massive stress on the worldwide health care structure. COVID-19 has been a terrible pandemic. TB, LO, and pneumonia both pose a serious risk of death [4,5]. Therefore, the prompt and correct diagnosis of these disorders is essential for providing effective care and sparing lives [6,7].

An LO on chest radiographs, which is frequently used by radiologists, basically refers to a white area of an unknown significance. Any object that prevents CRIs from passing through will appear white on a CRI because the lungs are typically dark on a CRI. Therefore, among other things, a white area among the normally black lungs could represent cancer, an infection, hemorrhage, fluid, or a foreign substance. The radiologist who reads the CRI makes an effort to provide an accurate and specific diagnosis using the medical data available, such as coughing and temperature, previous investigations, and laboratory results. So, if a patient visits the emergency room with a cough and fever, pneumonia will probably be found as an opacity on a CRI. An opacity that is more rounded in a long-term smoker is much more probable to be cancerous. White opacities in both lungs of a person with heart failure are most likely caused by edema or fluid in the lungs.

Globally, pneumonia is thought to be the main factor of child fatalities. A lung infection known as pneumonia can be brought on by either bacteria or viruses. Fortunately, antibiotics and antiviral medications work well in the treatment of this bacterial or viral infectious condition. However, the quicker identification of viral or BP and the subsequent administration of the appropriate medication can considerably aid in preventing a patient's health from deteriorating, which ultimately results in mortality [8]. Different kinds of pneumonia have been identified using CRIs, CT scans, and complete blood count (CBC) tests. Another kind of pneumonia affected by a ronavirus-2 is COVID-19. COVID-19, which now ranks as the largest pandemic in history, causes acute respiratory infections in humans. The virus initially infected people in China (Wuhan) in December 2019 [9]. Due to its rapid spread, COVID-19 is fatal to people. According to the WHO, there have been 761,402,282 confirmed COVID-19 cases reported globally to date, with 6,887,000 fatalities [10]. According to the WHO, there have been reported instances of COVID-19 in America, Europe, Africa, and Southeast Asia, respectively, of 268,252,496, 184,161,028, 60,719,433 and 9,431,508 cases. The Pakistani government reports that 1,518,083 COVID-19 cases have been documented there, with 30,304 deaths and 1,469,930 recoveries [11]. COVID-19 is typically detected with an antibody and PCR (polymerase chain reaction) test all over the world. These COVID-19 identification techniques are laborious and inefficient, and they take a while to receive results. So, with the aid of a doctor, chest radiology procedures such as CT scans and CRIs are performed to obtain outcomes more quickly. The signs of both illnesses include sneezing, coughing, fever, shortness of breath, and fatigue.

Furthermore, millions of individuals lose their lives to TB each year because it is a serious infection that primarily attacks the lungs. With timely detection and proper classification from other conditions with comparable radiologic features, TB can be treated to lessen the disease burden. With a worldwide death amount of approximately 1.8 million individuals and 10.4 million additional cases of human immunodeficiency virus (HIV) every year, the second most frequent reason for infectious illness deaths is tuberculosis, according to the WHO. Many underdeveloped countries are witnessing an increase in TB cases. Although both women and men can be affected, it seems to affect men more often. A lengthy course of antibiotic therapy and treatment is provided to patients with active TB [12]. Chest radiography has been recommended by the WHO and other organizations as an efficient approach for effective case discovery and existence examinations for the identification of TB.

All of the aforementioned illnesses share indications such as cough, sneezing, temperature, shortness of breath, and exhaustion. These lung disorders are categorized and identified utilizing CBC tests, RT-PCR, ultrasounds, and TST tests. These tests might take longer and still miss 20% of cases because the RT-PCR test has only an 80% sensitivity. After

24 h, a CT scan and a CRI were conducted in order to effectively control the false negatives in both asymptomatic and symptomatic individuals. A big issue with CT scans and chest radiographs, though, is the potential for COVID-19, pneumonia, LO, and TB diagnoses to be made at the same time. Moreover, manual tests are time-consuming and costly. To solve this, we require an efficient method that quickly and accurately categorizes CRIs employing trained convolutional neural networks (CNNs). Because they are less expensive, offer clean air sacs, and process more quickly than CT scans, CRIs are utilized frequently.

According to recent studies, DL-based artificial intelligence (AI) approaches can accurately diagnose a variety of disorders using CRIs with a level of precision comparable to that of experienced radiologists [13,14]. In resource-constrained situations when qualified radiologists are not easily accessible, these computer-aided detection (CAD) systems can increase practitioners' CRIs inter-reader variability and interpretation accuracy [15]. Similar to this, it has been shown that CAD approaches based on DL or conventional machine learning (ML), which might be utilized in clinical settings, can reliably categorize COVID-19 and other lung infections on chest radiographs [16,17]. A lot of novel DL architectures are created by researchers to identify different diseases using CRIs. For identifying COVID-19 using CRIs, the authors of [18] presented a novel COVID-Net DL framework and a COVIDx (easily accessible COVID-19 dataset). CRIs can be categorized by COVID-Net into any of the three categories. The framework consisted of two phases of projections, expansions, depthwise representations, and extensions, all relying on lightweight residual projection–expansion–projection–extension process models. The innovative TB-Net, a self-attention DL network for TB detection employing CRIs, was developed by the researchers in [19]. A highly specialized DL network containing attention condensers was called the TB-Net. They also tested TB-Net's decision-making abilities using an explainability-driven effectiveness verification process. In [20], the authors created a COVID-CXDNetV2 model for pneumonia and COVID-19 identification using CRIs. The model was based on ResNets and YOLOv2 architectures. Furthermore, in order to select the inputs (images or clinical data) and objectives of the system that could help obtain a trustworthy DL-based tool for difficulties related to COVID-19, the most pertinent and current medical studies and articles were examined in [21]. However, DL approaches use unstructured data in contrast to traditional ML methods, robotically extract robust traits, and generate reliable outcomes. Here are some advantages of DL over ML and other categorization techniques. This might lead to an increased accuracy with a bigger dataset. It will be quick and effective to evaluate and classify. It is not necessary to manually choose and extract features. This will be handled for you by a CNN. An unstructured, unclassified dataset is utilized for this procedure. DL makes it simple to build frameworks that create more precise outcomes in identifying and predicting particular lung diseases using chest radiographs.

According to our understanding, there are some drawbacks to current lung disease classification research: most (majority) previous studies utilized datasets with fewer images, (small datasets) or used the images of one dataset, which limits the generalization ability of the models. Less training data are available, the DL-based models are not completely generalizable, and the chances of overfitting are high. The vast majority of research uses transfer learning (TL) and conventional ML methods to detect lung diseases. Yet, the major issue with conventional ML (such support vector machines, or SVMs) is the extended training time for large datasets. The most significant restrictions in TL systems, however, are overfitting and negative transfer. One of its drawbacks is that pre-trained classification systems are frequently honed using the ImageNet database, which contains images unrelated to medical imagery. Moreover, pre-trained TL models require a lot of computing effort. It is still difficult to set up an effective CADS to promptly and effectively diagnose lung illness using chest radiographs. Furthermore, numerous researchers have suggested categorizing COVID-19, different kinds of pneumonia, TB, and standard CRIs. To the best of our knowledge, there is no single model and single dataset for lung disease classification into COVID-19, LO, pneumonia (or VP and BP), TB, and images of healthy

individuals. This motivates the development of an automatic and reliable model for the classification of various lung illnesses.

The DeepLungNet DL model is suggested as a solution to these constraints. It makes use of feature extraction that is based on filters, which can aid in obtaining an excellent classification performance. DeepLungNet extract features hierarchically and is capable of producing end-to-end learning, in contrast to traditional methods for feature extraction and selection that demand specialized knowledge. The convolutional layer and Leaky ReLU (LReLU) activation functions utilized to create the proposed framework extract the utmost important and in-depth features from the CRIs. The framework can minimize a range of weight characteristics by using a max-pooling procedure. We added batch normalization (BN) operations, convolutional layers, and group convolutional layers, a squeeze ConV layer with numerous $1 \times 1$-filter layers, and a combination of $1 \times 1$ and $3 \times 3$ ConV layers (expand layer) to make the suggested model a novel lung disease classification technique. Our approach is cost-effective, inexpensive, and less time consuming compared to traditional lung disease detection and classification approaches. Additionally, using the common Kaggle datasets that are open to the public, the proposed architecture was verified. Finally, the performance of our framework was compared with hybrid approaches (DL-based model plus SVM). To further show the model's usefulness, the suggested model was evaluated on a different publicly accessible dataset from the agriculture domain. The proposed structure works admirably in test accuracy for lung disease classification, according to the results. The following is a summary of the study's primary contributions:

- For the purpose of lung disease classification utilizing chest radiographs, an effective DeepLungNet model is proposed.
- Five-class classifications are made of CRIs into TB, Pneumonia, COVID-19, LO, and normal.
- Six-class classifications are made of CRIs into VP, BP, COVID-19, normal, TB, and LO.
- To improve the model's performance, demonstrate the model's generalizability, and prevent the overfitting issue, data augmentation is used.
- To determine the effectiveness of the DeepLungNet framework, we used hybrid methodologies to assess the classification performance of the presented approach on the similar experimental setting and dataset. For this goal, we employed a range of categorization criteria, including precision, f1-score, recall, and accuracy.
- The proposed framework is validated on another publicly accessible dataset from the agriculture domain to prove the generalization ability and usefulness of the framework.

The remainder of this article is structured as follows: Section 2 provides details about related work, Section 3 considers our used method, Section 4 describes the experiments and model's results, Section 5 hosts a discussion, and conclusions are made in Section 6.

## 2. Related Work

In the majority of nations, CRIs are routinely employed as a feasible choice for the identification of COVID-19 and other lung diseases. However, detecting COVID-19 is a challenging method that requires the clinical imaging of individuals. Lung cancer (LC) is one of the main reasons why people die. A prompt diagnosis may increase human survival. Image processing and ML have demonstrated significant potential for the analysis of pulmonary illnesses. To detect and categorize lung disorders, an issue that is still being studied and deserves more attention, a number of hybrid, ML, and DL methods have been published in the past. In-depth analyses of the DL approaches for LO, TB, COVID-19, VP, pneumonia, and BP are included in this section.

The authors of [22] used SVM and multi-level thresholding for COVID-19 detection or identification. The authors enhanced the contrast of input CRI by employing a median filter after examining the patient's CRIs. After that, a multi-level picture segmentation threshold is applied utilizing the Otsu objective function. After that the SVM was employed to distinguish between lungs with an infection and lungs without an infection. In [23], the

author presented a method based on autoregressive integrated moving average and least-squares SVM (LS-SVM) to identify or detect COVID-19 (ARIMA). The five countries with the maximum number of COVID-19 patients that have been confirmed are Italy, the United States, Spain, France, and the United Kingdom. The method used the verified cases as an input to forecast the disease's transmission one month in advance. For accuracy, LS-SVM surpassed ARIMA. A novel COVID-19 detection procedure built on a self-organization map and locality-weighted learning was proposed by the authors in [24]. (LWL-SOM). They utilized the SOM technique to gather the CRIs images into clusters on the basis of the same features in order to differentiate between healthy and COVID-19 patients. Furthermore, the LWL technique was utilized to develop a framework for recognizing COVID-19. The recommended framework enhanced the performance results for the correlation coefficients between normal and COVID-19 and pneumonia and COVID-19 cases. The present ML-based techniques that utilize AI assessment measures to differentiate between normal and COVID-19 patients outperform the suggested framework.

Unfortunately, standard ML approaches underperform DL approaches since they significantly trust human feature extraction and precise feature selection. DL approaches extract more robust deep features, make use of unstructured data, and make more precise outcomes compared to traditional ML algorithms. Nowadays, it has become standard procedure to automatically extract classification features using DL algorithms. Classifiers built on DL can be used to fully and automatically detect COVID-19 from CRIs.

For the categorization of CRIs, in [25], the authors proposed a DL framework with nine layers. The two-class classification of three illness categories, i.e., TB, pneumonia, and COVID-19, was accomplished by the means of six diverse datasets obtained from publicly accessible CRIs employing a DL framework that was completely trained from scratch. In [26], the authors trained a DL model with 6587 CRIs using stochastic gradient descent. The model successfully classified CRIs into four classes (normal, TB, pneumonia, and COVID-19) using 128 × 128 CRIs. In [27], the authors developed TL with VGG16 for TB diagnosis on CRIs. They refined the model using 1324 CRIs, and it produced satisfactory classification results for TB and healthy CXR images. In [28], the authors used a pre-trained DCNN-based Inception-V3 framework with TL. The collected dataset had 3532 CRIs in total, each of which were improved and scaled to 299 × 299. However, the study did not categorize TB in CRIs. In order to categorize CRIs, the authors of [29] combined VGG16 and attention mechanism. The techniques used to classify CRIs into COVID-19, normal, no findings, BP, and VP achieved a good classification performance on three CRI datasets.

Similar to this, in [30], the authors compared seven different popular DL neural network topologies. The small dataset employed in the study consisted of 50 CRIs and 25 CRIs from each of the COVID-19 and healthy patients. Only the classifier underwent training utilizing radiography; all other models underwent pre-training using the ImageNet database, which comprises about 14 million images of diverse types and is a broad image dataset. The best-performing designs in their tests were the VGG19. A similar approach was used to offer a modification of the VGG model that incorporates the convolutional COVID-19 block in [31]. The framework was assessed utilizing a diverse dataset consisting of 1887 images from 2 distinct publicly accessible datasets. Three categories of photographs were created: normal (654), pneumonia (864), and COVID-19 (300 images). In [32], many chest x-ray photographs from diverse sources were combined to form one of the main freely accessible collections of CRIs. Last, COVID-CXNet was created by the authors of [32] utilizing the TL approach and the well-known CheXNet model. This reliable model was able to recognize new COVID-19 pneumonia founded on important and relevant features with an accurate localization. In [33], the authors classified CRIs as belonging to COVID-19 and healthy people or VP patients using eleven CNN models. They considered three possible approaches to improving the COVID-19 identification designs by including extra layers. The models under examination were all well-known frameworks that have proved to be effective in applications for image recognition and detection. Using a COVID-19 radiography database, the recommended techniques for each explored design were

assessed, with the Xception and EfficientNetB4 models producing the best performance results. Moreover, the authors of [34] proposed a CNN-based architecture for COVID-19 detection from CRIs, increasing the test's efficacy and reliability. The suggested method combines a custom model with a TL approach to increase accuracy. Many pre-trained DL networks, including MobileNetV2, InceptionV3, VGG16, and ResNet50, were used to extract features. The performance indicators in this study were categorization and classification accuracy. The results of this research demonstrate that DL can identify COVID-19 in CRIs. InceptionV3 has attained the highest level of accuracy compared to other TL methods.

Previous studies have also used hybrid approaches, which integrate both DL- and ML-based procedures, in addition to ML approaches and DL models. In [35], the authors used a hybrid technique (SVM and deep-feature-based approach) to use CRIs to identify patients who were infected with COVID-19. SVM is utilized for classification instead of a DL-based framework since DL models require a sizable amount of training data. For COVID-19 categorization and classification, deep features from the fully connected (FC) layers of DL models are gathered and input into the SVM. Pneumonia, the norm, and COVID-19 are the distant CRIs data sources employed in the technique. The method helps doctors differentiate among normal, pneumonia, and COVID-19 cases. The characteristics of 13 DL frameworks were used to assess the SVM algorithm's COVID-19 identification performance. Resnet50 and SVM attained the highest classification performance. Furthermore, in [36], the authors used CRI data to train CNN frameworks as feature extractors and the SVM as a classification algorithm to assess whether the individuals were healthy, had pneumonia, or were suffering from COVID-19. The tests compared various classes, feature extraction frameworks, feature selection algorithms, and kernels. To discriminate among the three groups of pneumonia, COVID-19, and normal, the investigators employed the resnet50, resnet18, resnet101, and GoogleNet TL methods. Using resnet101, resnet50, resnet18, and GoogleNet, they were able to achieve the highest average accuracy.

The previous works could be expanded much more. According to the aforementioned literature review, different ML, DL, and hybrid techniques were used to classify various lung illnesses based on CRIs. However, existing approaches are unable to classify lung diseases into TB, VP, pneumonia, BP, COVID-19, and LO. Additionally, to evaluate the generalizability and robustness of models, we need to train and test models on multiple datasets or datasets with images from multiple datasets. The majority of studies employed only one dataset for model performance validation. This paper proposes a DeepLungNet model which is trained on the images from multiple datasets to verify the robustness of the model. This study's main objective is to detect multiple lung diseases using a single model with an adequate accuracy while minimizing false positives. Analysis of the data reveals that the suggested system for lung disease classification is useful and reliable.

## 3. Methodology

The application of DL approaches has already had a significant positive impact on the fields of image processing (more specifically, medical imaging). In this study, we suggested the DeepLungNet DL framework for lung disease classification using chest radiographs. Using (our integrated) dataset, we will categorize chest radiographs into the following four groups: TB, normal, LO, COVID-19, and pneumonia. Figure 1 depicts the suggested strategy's abstract representation. To put the suggested technique into practice, we provided the model images of chest radiographs. The input images for the datasets were a variety of sizes. Then, we used pre-processing to shrink the dimensions of the input images to 224-by-224 pixels in order to assure homogeneity and speed up the procedure. To further categorize CRIs into the 5 ideal configurations, a DeepLungNet architecture with only 20 convolutional layers was created. For each experiment, independent datasets were used for testing and training. We specifically utilized 80% of the dataset for training, whereas we used 20% for testing purposes. The two datasets were then used to evaluate the proposed model.

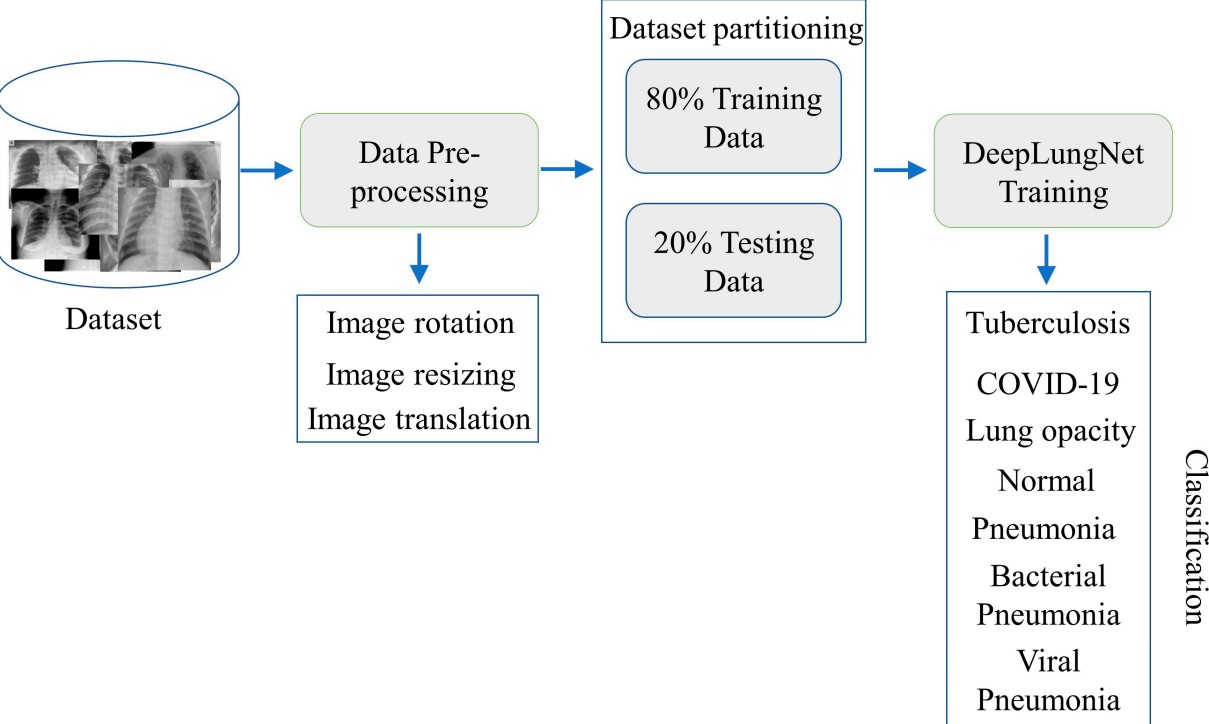

**Figure 1.** General workflow of the proposed method.

### 3.1. Data Pre-Processing

#### 3.1.1. Data Augmentation

One of the issues when attempting to use DL frameworks for medical imaging detection and classification tasks is the lack of suitable data (balanced data) to train the DL frameworks. It is necessary to collect more medical imaging data, yet doing so requires a large amount of time and money. By applying data augmentation strategies to the pre-existing data without gathering any new medical imaging data, we boosted the amount of data that are now available; we used data augmentation techniques to overcome the class imbalance issue. The radiograph scans in the dataset were randomly rotated at an arbitrary angle between −20 and 20 degrees and moved up to 30 pixels in both the vertical and horizontal directions. To create new images, we translated the existing images at random between 0.9 and 1.1. It is critical to keep in mind that in each training session, the imageDataAugmenter function continuously produces sets of enhanced images. By dramatically increasing the dataset's image count, we were able to train our deep learning framework with more training images and improve its performance.

#### 3.1.2. Image Resizing

The datasets input CRIs come in a variety of dimensions. To ensure homogeneity and improve the processing speed, we pre-processed the radiographs to scale them to $224 \times 224$ pixels in accordance with the requirements (input picture) of our model.

### 3.2. Dataset Partitioning

The CRIs were separated into testing and training groups for each experiment. More exactly, the framework was trained on 80% of the dataset, then tested on the remaining 20%.

### 3.3. Deep DeepLungNet Architecture Details

In this study, we proposed the DeepLungNet framework for lung illness classification. Only 20 learnt layers, i.e., 18 convolutional layers and 2 FC layers, make up the DeepLungNet model. In total, there are 64 layers in our architecture: 1 for the picture input, 16 for convolutions, 2 for group convolutions, 18 for batch normalization (BN), 19 for leaky *ReLU* (LReLU), 1 for maximum pooling, 2 for fully connected, 1 for average pooling, 1 for dropout, 1 for softmax, and 1 for classification. The leaky *ReLU* activation function comes after the ConV and group convolutional layers.

Table 1 displays the DeepLungNet model's architecture. In the DeepLungNet model, the first (input) layer is the top (initial) layer. Its size is equivalent to the size of the input features, and it contains I × J elements. For processing, our framework takes input images with a 224 × 224-pixel size. ConV layers with a kernel size of 7 × 7, 3 × 3, and 1 × 1 are used, which performs a ConV operation to create feature maps. The first ConV layer extracts the feature from the CRIs (of size 224 × 224) by using 64 filters of size 7 × 7 with a shift of 2 × 2 and padding of 3 × 3. Following the use of convolutions and kernel, the output of the ConV layers (feature map) can be derived by using Equation (1). Equation (1) represents the ConV operation between the image and kernel [34]:

$$f_c^k(m,n) = \sum_d \sum J_d(r,s).i_c^k(v,w) \tag{1}$$

**Table 1.** DeepLungNet architecture details.

| S No | Operation | Layers | Kernel | No of Filters | Padding | Stride |
|------|-----------|--------|--------|---------------|---------|--------|
| 1 | | Input | | | | |
| 2 | ConV | ConV (BN, LReLU) | 7 × 7 | 64 | 3 × 3 | 2 × 2 |
| | | ConV (BN, LReLU) | 1 × 1 | 16 | | |
| 3 | Fire module | ConV (BN, LReLU) | 3 × 3 | 64 | 1 × 1 | |
| | | ConV (BN, LReLU) | 1 × 1 | 64 | | |
| 4 | Pooling | Max-Pooling | 3 × 3 | | 1 × 1 | 2 × 2 |
| 5 | ConV | Group ConV (LReLU, CCN) | 5 × 5 | 128 | [2 2 2 2] | |
| 6 | ConV | ConV (BN, LReLU) | 3 × 3 | 64 | 1 × 1 | |
| 7 | ConV | ConV (BN, LReLU) | 3 × 3 | 64 | 1 × 1 | |
| 8 | ConV | ConV (BN, LReLU) | 3 × 3 | 128 | 1 × 1 | 2 × 2 |
| 9 | ConV | ConV (BN + LR) | 1 × 1 | 128 | | 2 × 2 |
| 10 | ConV | ConV (BN + LR) | 1 × 1 | 256 | | 2 × 2 |
| 11 | ConV | ConV (BN + LR) | 3 × 3 | 256 | 1 × 1 | 2 × 2 |
| 12 | ConV | ConV (BN + LR) | 3 × 3 | 512 | 1 × 1 | 2 × 2 |
| 13 | ConV | ConV (BN + LR) | 1 × 1 | 512 | | 2 × 2 |
| 14 | ConV | ConV (BN + LR) | 3 × 3 | 32 | Same | 2 × 2 |
| 15 | ConV | Group ConV (BN + LR) | 3 × 3 | 32 | Same | |
| 16 | ConV | ConV (BN + LR) | 1 × 1 | 16 | Same | |
| 17 | ConV | ConV (BN + LR) | 1 × 1 | 96 | Same | |
| 18 | ConV | Group ConV (BN + LR) | 3 × 3 | 96 | Same | 2 × 2 |
| 19 | ConV | ConV (BN + LR) | 1 × 1 | 24 | Same | |
| 13 | | FC + LReLU + dropout | | | | |
| 15 | | Average pooling + FC + softmax + classification | | | | |

$f_c^k$ represents the output feature map, and $j_d$ (r, s) represents the chest radiographs which are multiplied by the $i_c^k(v, w)$ index of the kth kernel of the cth layer. After employing convolutions on the input chest radiographs, the output of size $o = ((i - k) + 2p)/(s + 1)$ is formed, whereby *i* stands for the input, *p* for padding, *k* for kernel size, and *s* for steps.

All ConV and group ConV layers are followed by activation functions. Following convolutional layers are the activation functions. The most popular activation functions in the past were sigmoid and tanh. These limitations led researchers to develop substitute activation functions, such as the rectified linear unit (ReLU) and its derivatives (ELU, Noisy ReLU, and LReLU), which are presently utilized in the bulk of DL applications. A node in a layer uses the activation function to transform the weighted sum of the input into the output. All neurons with negative values are deactivated by the *ReLU* activation function, rendering a significant percentage of the framework (network) indolent. To enhance the

model's classification performance, we applied an enhanced *ReLU* activation function (LReLU activation function) to describe the *ReLU* activation function as a very minor linear percentage of x rather than stating that it be 0 for negative input values. Here is how this activation function was intended: the LReLU, in contrast to *ReLU*, does not deactivate the inputs and also generates an output for negative values. The LReLU activation function works according to Equation (2):

$$f(x) = \max(0.01 \times x, \ x) \tag{2}$$

When given a positive input, the LReLU function returns $x$, but when given a negative input, it returns 0.01 times $x$ (small value).

To normalize the outputs of ConV layers, we used the BN operation. BN enables regularization and accelerates the learning process of neural networks, and it also helps to prevent overfitting.

The output feature of the first ConV layer is delivered into the next convolutional layer (fire module) after employing the activation function (LRelU) and BN operation. Three ConV layers make up the Fire module: a squeeze ConV layer with numerous $1 \times 1$-filter layers, then $3 \times 3$ and $1 \times 1$ ConV layers (expand layer). To decrease the total parameters, we chose $1 \times 1$ layers. The number of input channels multiplied by the number of filters and the filter size, which is three, yields the total number of parameters in the layer. Therefore, we utilized fewer kernels in the squeeze layer than in the expand layer to reduce the number of inputs to $3 \times 3$ kernels. We used the padding of 1 pixel in the ConVlayers with $3 \times 3$ filters in order to make the output of the $3 \times 3$ and $1 \times 1$ filters the same size. After the fire module, we employed a maximum pooling layer. The maximum pooling layers with a stride of $2 \times 2$ after the fourth convolutional layer were used for down-sampling. The spatial size, computational complexity, the number of parameters, and calculations were all reduced by this layer. Equation (3) shows the working of the maximum pooling layer.

$$f(x) = \{x_1, \ x_2, \ x_3, \dots, \ x_k\} \tag{3}$$

The $f(x)$ represents the optimal feature map. In our model, a filter size of $3 \times 3$ and a stride of $2 \times 2$ is utilized to select the highest value from the neighboring pixels (in a radiograph image) using maximum pooling.

The output of the fire module is passed as an input to the ConV layer taking, 64 kernels of size $3 \times 3$ and padding $1 \times 1$. Similarly, the next ConV layer also applies the 64 kernels of size $3 \times 3$ with padding of $1 \times 1$. The activation function after this convolutional layer comes after the additional layer. We applied the activation after the addition layer. The next six convolutional layers are connected using shortcut connections, whereas the remaining (last) six convolutional layers are connected sequentially.

The first FC layer receives the output of the final (i.e., eighteenth) ConV layer. A one-dimensional feature vector is created from the two-dimensional feature map that was taken from the ConV layers by the FC layer. The operations of a FC layer are elaborated in Equation (4).

$$a_i = \sum_{j=0}^{m \times n - 1} w_{ij} \times x_i + b_i \tag{4}$$

where $i$, $m$, $n$, $d$, $w$, and $b$ stand for the output index, width, height, depth, weights, and bias of the FC layer, respectively. We used the dropout layer after the initial FC (to prevent overfitting). The final FC layer is followed by the softmax and classification layers.

### 3.4. Hyper-Parameters

The choice of hyper-parameters is central to the effectiveness of DL architectures. In order to discover the appropriate value for each hyper-parameter given the wide range of alternatives available, we investigated the effectiveness of the suggested DeepLungNet model using a number of hyper-parameter settings. We choose a few hyperparameters for

a model to determine how the DL architecture hyperparameter affects the representation of the entire network. The model is trained on dataset 1 using different parameters, and the model performance metrics are examined. Until the model has reached optimal accuracy, as shown in Table 2, this process is repeated using a new set of values for hyperparameters. Table 3 shows the final hyper-parameter values. We employed the stochastic gradient descent optimization approach since it is effective for larger datasets, rapid, and memory efficiency. In order to account for the possibility of overfitting, we trained the model for 50 epochs.

**Table 2.** Hyperparameters tuning results.

| Experiment No | Learning Rate | Epochs | Dropout | Accuracy |
|:---:|:---:|:---:|:---:|:---:|
| 1 | 0.1 | 30 | 0.5 | 96.89 |
| 2 | 0.5 | 35 | 0.4 | 97.12 |
| 3 | 0.01 | 30 | 0.2 | 97.26 |
| 4 | 0.05 | 35 | 0.6 | 97.0 |
| 5 | 0.001 | 40 | 0.5 | 97.47 |

**Table 3.** Hyperparameters of proposed architecture.

| Parameter | Value |
|:---:|:---:|
| Learning rate | 0.001 |
| Optimization algorithm | SGDM |
| Validation frequency | 30 |
| Verbose | false |
| Activation Function | LReLU |
| Test Size | 0.2 |
| Train Size | 0.8 |
| Dropout | 0.5 |
| Iterations per epoch | 42 |
| Shuffle | Every epoch |
| Maximum Epochs | 40 |

## 4. Results

This segment thoroughly examines the outcomes of the several tests conducted to gauge how well our model works. We outline the experimental strategy and performance metrics we used to evaluate the efficacy of our strategy. Further information about the datasets is also provided in this section. We made use of publicly accessible Kaggle datasets to evaluate the efficacy of our strategy.

### 4.1. Datasets

We tested the usefulness and robustness of our proposed approach by using images from multiple datasets and created two integrated datasets and evaluated our model on the integrated datasets. The CRIs of the datasets are diverse in terms of illumination conditions, format, angles, dimensions, bit depth, and size, etc. The datasets contain PNG and JPG images of different resolutions. All images of the datasets are grayscales, and the bit depth of all the images in the datasets is eight.

#### 4.1.1. Dataset 1

We created a dataset combining CRIs from publicly available datasets since there was a lack of a standard dataset for classifying lung illnesses (normal, TB, COVID-19, LO, or pneumonia). To create our own integrated dataset for five-class classifications, we have used the COVID-19 and LO images of the standard "COVID-19 Chest Radiography Database" dataset [37]. This dataset contains four types of chest radiographs, i.e., COVID-19, LO, Pneumonia, and normal. We have utilized pneumonia and normal CRIs of the publicly available "Chest X-ray (COVID-19 & Pneumonia)" dataset [38]. Furthermore,

we have used TB CRIs of the "Lung Disease Dataset (4 types)" dataset [39]. This dataset contains images of four types of lung diseases (i.e., VP, coronavirus disease, BP, TB) along with normal CRIs. The details of the dataset are provided in Table 4.

**Table 4.** Details of the datasets used.

| Dataset | Training Images | Testing Images | Total Images |
|---------|-----------------|----------------|--------------|
| Dataset 1 | 11,277 | 2819 | 14,096 |
| Dataset 2 | 9639 | 2410 | 12,049 |

### 4.1.2. Dataset 2

We have created another dataset by combining the CRIs from publicly available datasets with different lung diseases, i.e., VP, BP, COVID-19, normal, TB, and LO. As there does not exist any single dataset which can be used to classify the aforementioned lung diseases, we have combined the images of three publicly available datasets and used that integrated dataset to validate the performance of our framework. More specifically, we used COVID-19 and LO images from the "COVID-19 Chest Radiography database" dataset [37]. We applied augmentation on all images of the "Chest X-ray (COVID-19 & Pneumonia)" dataset [38] and used normal chest radiographs. Furthermore, we have used TB, BP, and VP CRIs of the "Lung Disease Dataset (4 types)" dataset [39]. The details of the dataset are provided in Table 4. Furthermore, Figure 2 shows some representative samples of dataset 2.

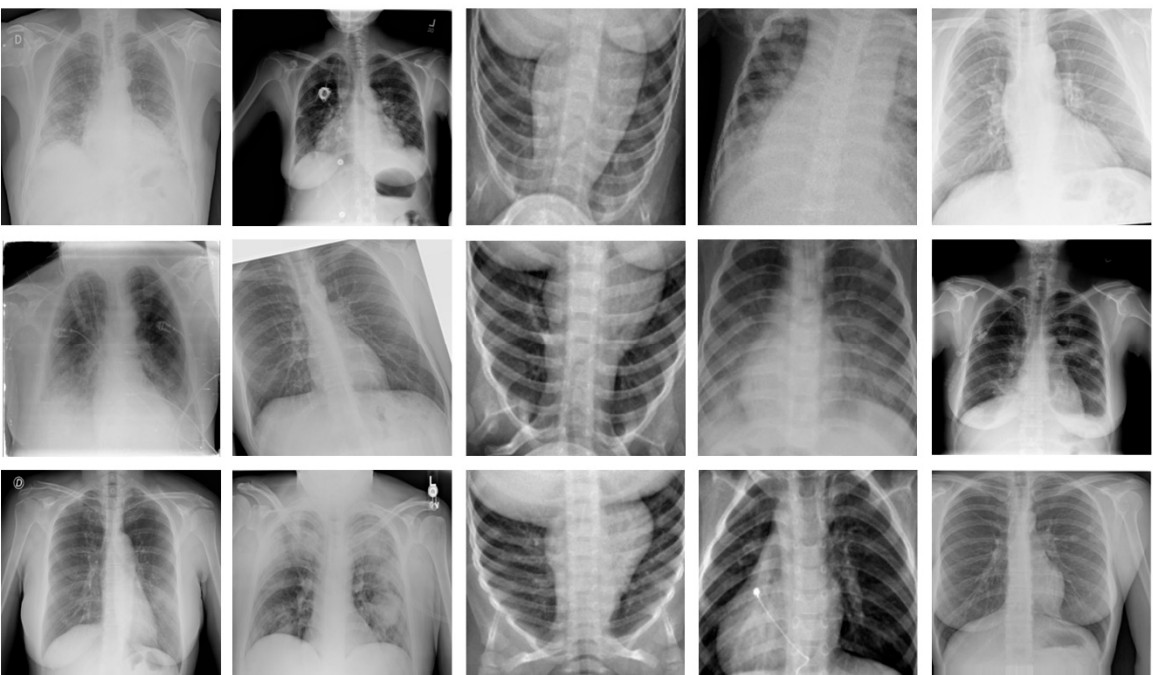

**Figure 2.** Representative samples from dataset 2: the first column shows COVID-19, the second column shows LO, the third column shows normal, the fourth column shows pneumonia, and the fifth column shows TB radiograph images.

### 4.2. Evaluation Metrics

We assessed the effectiveness of our system using the metrics for precision, sensitivity, accuracy, and F1-score. The following is the formula for these measures:

$$Accuracy = (TN + TP)/TS \tag{5}$$

$$Precision = \frac{TP}{TP + FP} \qquad (6)$$

$$Sensitivity\ (recall) = \frac{TP}{TP + FN} \qquad (7)$$

$$F1\_score = 2 \cdot \frac{Precision \times Recall}{Precision + Recall} \qquad (8)$$

where *TP*, *TN*, *TS*, *FN*, and *FP* denote the true positive, true negative, total samples, false negative, and false positive, respectively.

### 4.3. Experimental Setup Ad Evaluation

All of the studies were carried out on a laptop with an Intel (R) Core (TM) i5-5200U CPU and 8 GB of RAM. The strategy was carried out using MATLAB R2020a. The datasets for training and testing were separated for each experiment. We performed several experiments to assess the classification performance of our proposed model for CRIs. Table 5 provides the details of the software and hardware utilized for the implementation of the proposed method.

**Table 5.** Details of software and hardware utilized for implementation.

| Sr. No | Name | Experiment Parameters |
|--------|------|------------------------|
| 1 | CPU | Intel (R) Core (TM) i5-5200U |
| 2 | Type of system | 64-bit, Windows 10 |
| 3 | RAM | 8 GB |
| 4 | ROM | 500 GB |
| 5 | Development tool | MATLAB R2020a |

4.3.1. Performance Evaluation on Dataset 1 (5-Class Lung DISEASE Classification)

The vital aim and goal of this experiment is to validate the multi-class classification ability of our proposed DeepLungNet model and to categorize CRIs into TB, pneumonia, COVID-19, LO, and normal. For this experiment, we combined the images of three publicly available datasets and used that integrated dataset to validate the performance of our model (dataset 1). More specifically, we used a total of 4000 CRIs from the "COVID-19 Chest Radiography database" dataset (2000 COVID-19, 2000 LO). We applied augmentation on all the images of "Chest X-ray (COVID-19 & Pneumonia)" and used 4420 normal and 4456 pneumonia chest radiographs in this experiment. Furthermore, we have used 1220 TB CRIs of the "Lung Disease Dataset (4 types)" dataset. We utilized 11,277 CRIs (1600 COVID-19, 976 TB, 1600 LO, 3536 normal, and 3565 pneumonia chest radiographs) to train our model. The outstanding 2819 CRIs (400 COVID-19, 244 TB, 400 LO, 884 normal, and 891 pneumonia chest radiographs) were utilized for testing our model. The training set for DeepLungNet architecture for lung disease classification is composed of the parameters specified in Table 3. The suggested model required 2383 min and 23 s for training for lung disease classification. However, this time is determined by the total number of iterations and epochs. Throughout the training phase, DeepLungNet underwent a total of 3520 iterations (88 iterations per epoch), with a total of 40 epochs. At epoch 50, the framework's average testing precision, accuracy, f-measure, and recall were 93.2%, 97.47%, 93.4%, and 93.6%. In Figure 3, accuracy and loss are shown so that you can see how well our framework is trained. The loss function reveals how fine the system can identify the CRIs in dataset. After epoch 27, our model's training accuracy and loss mostly remain the same, whereas the model's testing accuracy basically remains the same after epoch 37, showing that it can classify lung diseases with a higher classification performance even at less epochs than 40.

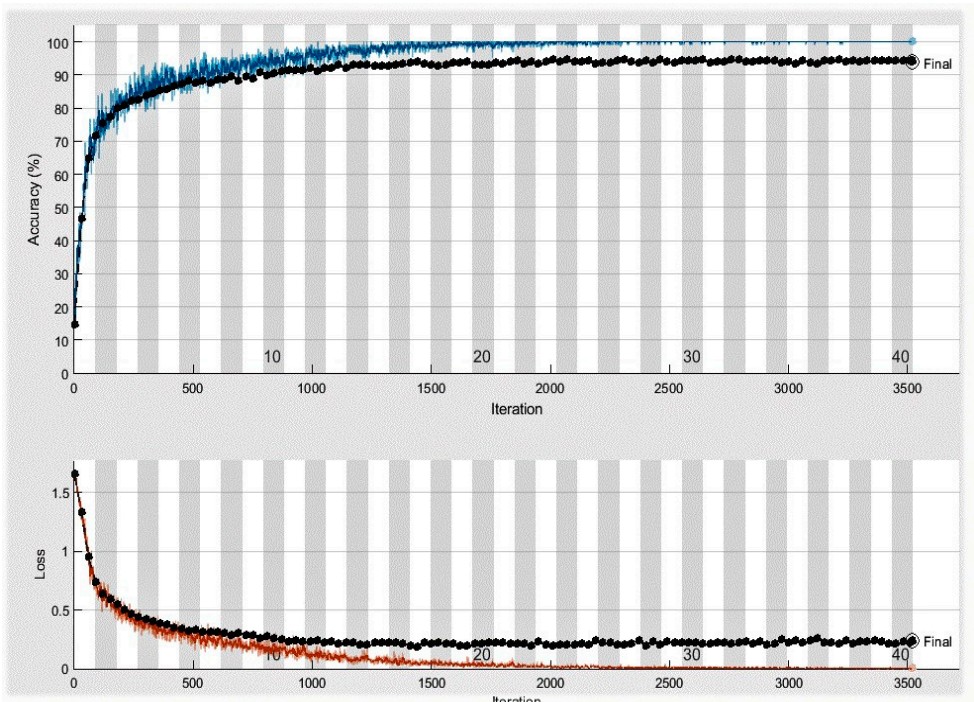

**Figure 3.** Testing and training loss and accuracy of proposed model (five-class lung disease classification).

The confusion matrix (CM) for our framework's lung disease classification testing phase is revealed in Table 6. The proposed design misclassified 160 CRIs out of 2819, including 29 COVID-19, 11 TB, 59 LO images, and 61 Pneumonia images. It is significant to mention that our framework classified all normal images correctly. The results show that our model (method) has higher TN and TP values and lower FN and FP values.

**Table 6.** CM obtained by the DeepLungNet framework (five-class lung disease classification).

| | | **Predicted Class** | | | | |
|---|---|---|---|---|---|---|
| | Disease Class | COVID-19 | TB | LO | Normal | Pneumonia |
| | COVID-19 | 371 | 1 | 27 | 0 | 1 |
| | TB | 2 | 233 | 9 | 0 | 0 |
| True class | LO | 42 | 9 | 341 | 2 | 6 |
| | Normal | 0 | 0 | 0 | 884 | 0 |
| | Pneumonia | 0 | 0 | 3 | 58 | 830 |

To evaluate the effectiveness and validity of the suggested approach, several chest radiograph images must be precisely identified and categorized. To do this, we assessed how well the offered technique classified each radiological image (i.e., TB, COVID-19, LO, pneumonia, or normal). Table 7 displays the recall, F1-score, precision, and accuracy results of the recommended strategy's class-wise radiograph classification performance. The suggested approach demonstrates a cutting-edge performance for each evaluation criterion, as verified in Table 5; the bar chart of the results are shown in Figure 4. According to the results, the majority of radiological images were identified accurately, producing the maximum level of accuracy. The strength of the recently established DL framework, which more properly replicates each class, is primarily responsible for the increased radiograph classification accuracy.

**Table 7.** Class-wise performance of the DeepLungNet framework.

| Class | N(Classified) | N(Truth) | Precision | Accuracy | Recall | F1-Score |
|---|---|---|---|---|---|---|
| COVID-19 | 400 | 415 | 93.0 | 97.41 | 89.0 | 91.0 |
| TB | 244 | 243 | 95.0 | 99.26 | 96.0 | 96.0 |
| LO | 400 | 380 | 85.0 | 96.52 | 90.0 | 87.0 |
| Normal | 884 | 944 | 100 | 97.87 | 94.0 | 97.0 |
| Pneumonia | 891 | 837 | 93.0 | 97.59 | 99.0 | 96.0 |

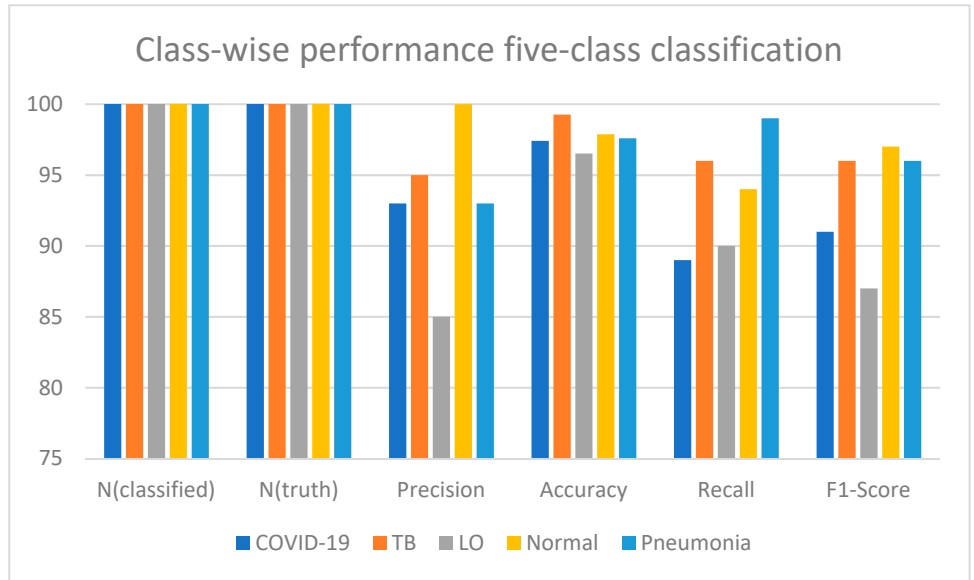

**Figure 4.** Bar chart of class-wise performance of five-class classification.

The suggested DeepLungNet's Receiver Operating Characteristic (ROC) curve, shown in Figure 5, expresses the lung disease classification performance of the DeepLungNet framework. We used the MATLAB function perfcurve to calculate the ROC. Threshold values were applied by the ROC to the outputs in the [0,1] range. For each threshold, the FP Ratio and TP Ratio were calculated. The FP to TP ratio is depicted on the ROC curve, illuminating the sensitivity of the classification model. The area under the curve (AUC) is a crucial assessment criterion for classifiers since it shows how dissimilar different categories are from one another; it establishes how well the model can differentiate between classes. The model more effectively distinguishes between different (i.e., COVID-19, LO, normal, pneumonia, or TB) individuals when the AUC value is near to one. This demonstrates a high level of competence. It is clear that DeepLungNet reported an AUC value of 0.9940. The proposed DeepLungNet framework was more accurate in classifying lung diseases because it is better at extracting distinctive features from CRIs. The batch normalization method of the suggested model standardizes, regularizes, and minimizes generalization error for each mini-batch of inputs to a layer.

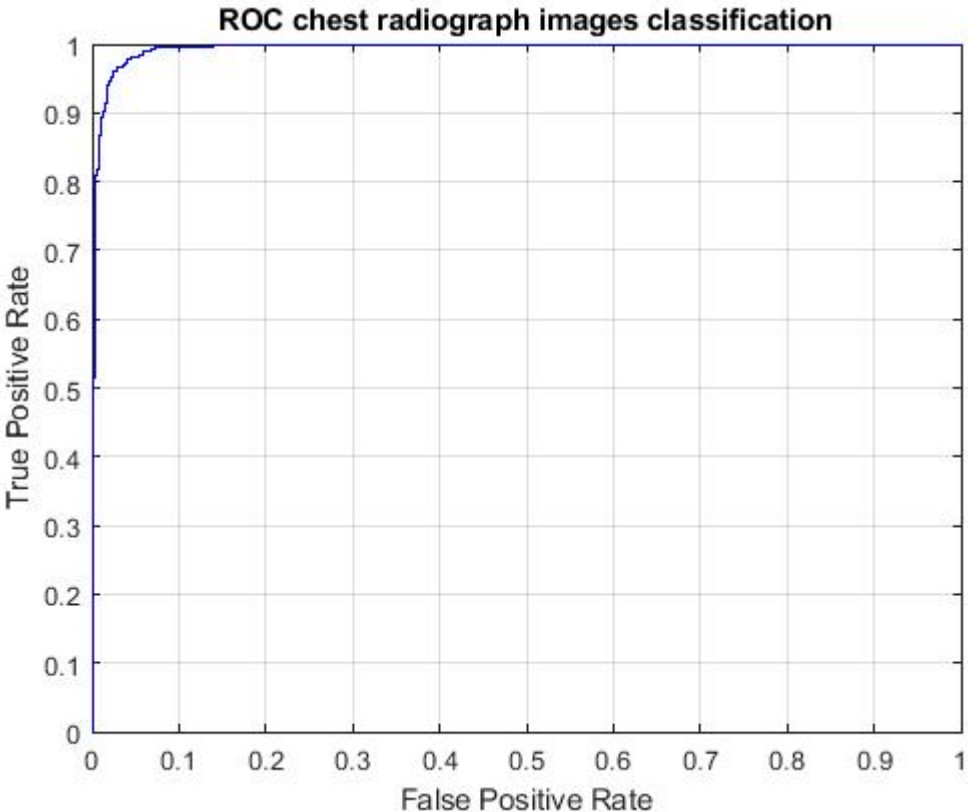

**Figure 5.** AUC plot of the proposed DeepLungNet model.

4.3.2. Performance Evaluation on Dataset 2 (Six-Class Lung Disease Classification)

To further evaluate and validate the reliability and performance of our DeepLungNet model, we have designed an experiment for (six-class classification) lung disease classification. We have used the proposed model to classify chest radiographs with different lung diseases, i.e., VP, BP, COVID-19, normal, TB, and LO. As there does not exist any single dataset which can be used to classify the aforementioned lung diseases, we have combined the images of three publicly available datasets and used that integrated dataset to assess the performance of our technique. More precisely, we used a total of 4000 CRIs from the "COVID-19 Chest Radiography database" dataset (2000 COVID-19, 2000 LO). We applied augmentation on all images of "Chest X-ray (COVID-19 & Pneumonia)" and used 4420 normal chest radiographs in this experiment. Furthermore, we have used 1220 TB, 1205 BP, and 1204 VP CRIs of the "Lung Disease Dataset (4 types)" dataset. We used 9639 images (1600 COVID-19, 976 TB, 1600 LO, 3536 normal, 964 BP, 963 VP, and 3565 pneumonia chest radiographs) to train our model. The outstanding 2410 images (400 COVID-19, 244 TB, 400 LO, 241 BP, 241 VP, and 884 normal chest radiographs) were utilized for testing our model. The training set for our DeepLungNet architecture for lung disease classification is composed of the parameters specified in Table 3. The suggested model required 1884 min and 43 s for training for lung disease classification into COVID-19, LO, BP, VP, TB, and normal. However, this time is determined by the total number of iterations and epochs. Throughout the training phase, DeepLungNet underwent a total of 3000 iterations (75 iterations per epoch), with a total of 40 epochs. The model's f-measure, average validation accuracy, precision, recall, and accuracy were 95.57%, 80.0%, 82.16%, and 81.06% at epoch 40, respectively. You can assess how well our framework is trained by looking at the accuracy and loss in Figure 6. Our model's testing accuracy essentially stays the same after epoch 37, although the model's training accuracy and loss essentially stay the same after epoch 27. This indicates that the model can classify lung diseases with the best accuracy even at epochs lower than 40.

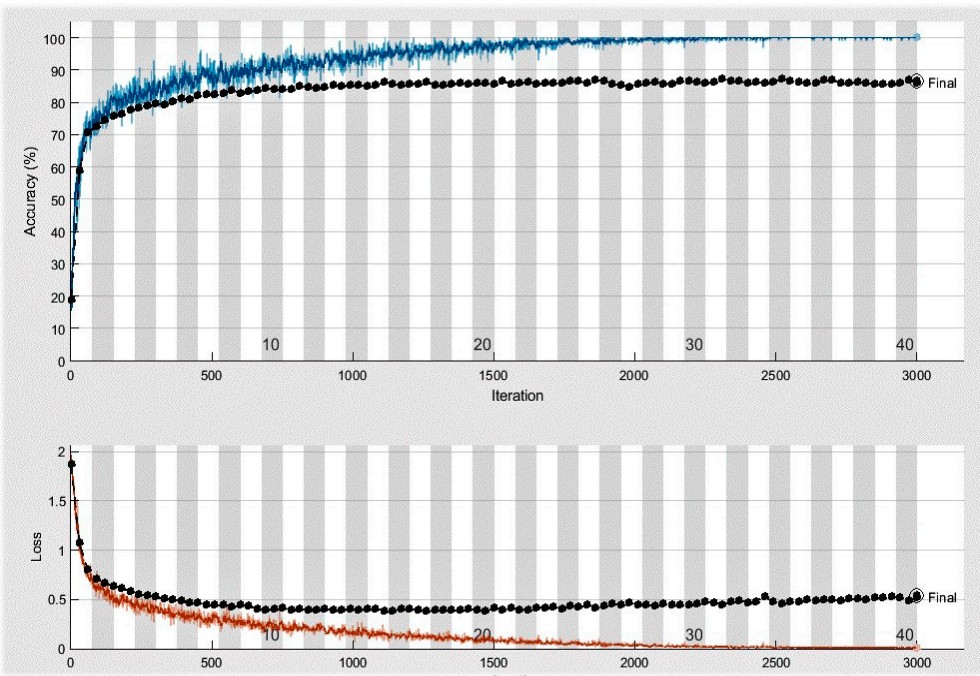

**Figure 6.** Testing and training loss and accuracy of the proposed model (six-class lung disease classification).

The CM for our framework's lung disease classification testing phase is revealed in Table 8. The proposed design misclassified 320 CRIs out of 2410, including 19 COVID-19, 151 TB, 57 LO images, 11 VP, and 82 BP images. It is vital to mention that our method classified all normal images correctly. The results show that our approach has higher TP and TN values and lower FP and FN values.

**Table 8.** CM obtained by the proposed model (six-class lung disease classification).

| | | **Predicted Class** | | | | | |
|---|---|---|---|---|---|---|---|
| | **Disease Class** | **TB** | **COVID-19** | **VP** | **BP** | **LO** | **Normal** |
| | TB | 93 | 2 | 15 | 114 | 9 | 11 |
| | COVID-19 | 1 | 381 | 5 | 0 | 13 | 0 |
| True class | VP | 0 | 4 | 230 | 0 | 7 | 0 |
| | BP | 22 | 7 | 17 | 159 | 18 | 18 |
| | LO | 2 | 49 | 5 | 1 | 343 | 0 |
| | Normal | 0 | 0 | 0 | 0 | 0 | 884 |

It is necessary to accurately identify and categorize multiple CRIs to assess the worth and validity of the presented strategy. To do this, we evaluated how well each radiological image was classified by the proposed technique (i.e., VP, BP, TB, COVID-19, LO, or normal). The class-wise radiograph classification performance of our new technique is shown in Table 9 in terms of the recall, accuracy, F1-score, and precision (Figure 7). As shown in Table 9, the suggested approach demonstrates a cutting-edge performance for each evaluation criterion. The improved radiograph classification accuracy is mostly attributable to the stability of the recently developed DL framework, which more accurately reflects each class.

**Table 9.** Class-wise performance of the proposed framework.

| Disease Class | N(Truth) | N(Classified) | Accuracy | Precision | Recall | F1-Score |
|---|---|---|---|---|---|---|
| TB | 118 | 244 | 92.7 | 38.0 | 79.0 | 51.0 |
| COVID-19 | 443 | 400 | 96.64 | 95.0 | 86.0 | 90.0 |
| VP | 272 | 241 | 97.8 | 95.0 | 85.0 | 90.0 |
| BP | 274 | 241 | 91.83 | 66.0 | 58.0 | 62.0 |
| LO | 390 | 400 | 95.68 | 86.0 | 88.0 | 87.0 |
| Normal | 913 | 884 | 98.8 | 100.0 | 97.0 | 98.0 |

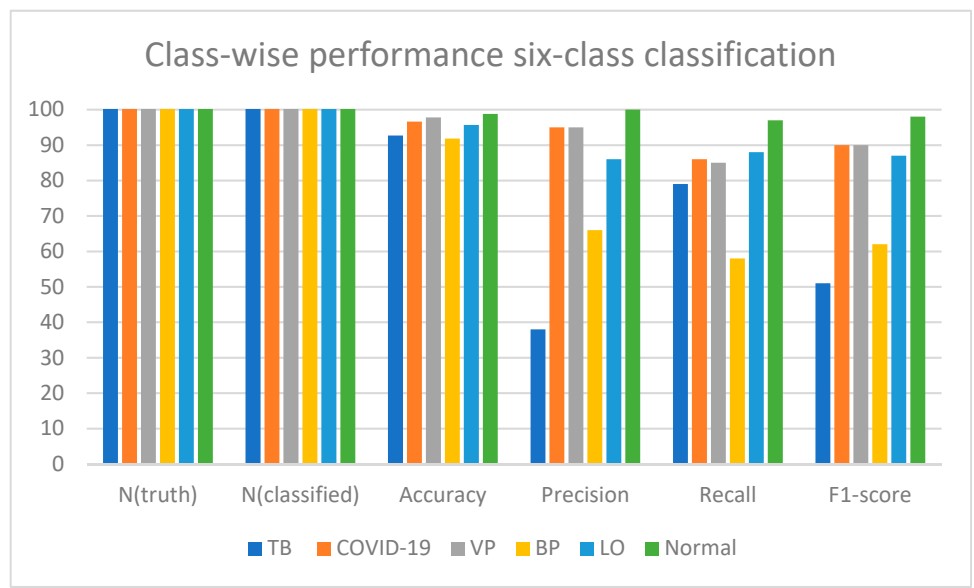

**Figure 7.** Bar chart of class-wise performance of six-class classification.

### 4.3.3. Performance Evaluation on Lemon Quality Dataset

Even though the experiments show that the DeepLungNet model performs well in the case of lung disease classification utilizing multiple datasets of CRIs, more testing on a different dataset in other domains is required to show that the proposed model is resilient, stable, and generally applicable. We performed an experiment to assess our DeepLungNet framework on another dataset for a different classification task. The aim of this experiment is to assess how well the suggested method performs in the agriculture domain. The dataset used for this purpose is the lemon quality dataset [40], which has been created to explore potential solutions to the fruit quality control problem. It has 2.533 images (300 × 300 pixels). On a concrete surface, photographs of lemons are taken. This surface's empty images are also included in the dataset. The dataset includes images of lemons of various qualities in various sizes and lighting settings (all in the daytime). The dataset contains 452 empty backgrounds, 951 bad-, and 1125 good-quality lemon images. We used 2076 (761 bad-quality and 900 good-quality) images in this experiment. We used 1661 images for the training of our model, whereas the remaining 415 (190 bad-quality and 225 good-quality) images were used for the testing of our model. We achieved a classification precision, accuracy, recall, and F1-score of 95.0%, 95.3%, 95.0%, and 95.0%, respectively. The accuracy of 95.3% demonstrates the usefulness and generalization power of our method for lemon image classification.

### 4.3.4. Comparison with Hybrid Approaches

The effectiveness of our DeepLungNet classifier is investigated in this part through the development of a hybrid experiment for CRI classification to identify COVID-19, Pneumonia, LO, and TB diseases. As we have achieved the greatest classification performance

in the case of five-class classifications, we have compared our approach with the hybrid approaches on dataset 1. It is emphasized that using an SVM for classification at the end of the model in place of a softmax will result in dramatically higher classification results. The most popular deep CNNs, such as Alexnet [41], Squeezenet [42], MobileNetv2 [43], Shufflenet [44], and GoogleNet [45], were applied to extract features in order to overcome this issue. We then used such values as inputs to construct a linear SVM as the decision function using these features. C and Gamma hyperparameter values were tuned to 1.0 and 0.1, correspondingly, to obtain the best results. These models require input images of various sizes and have multiple layers. While the densenet201 framework has 201 layers and processes input CRIs with a size of 224 × 224, the Squeezenet framework has 18 layers and processes an image with a size of 227 × 227. Therefore, employing enhanced image datastore functionality, we scaled the chest radiographs to match the input picture specifications of these models. We trained these DL-based models using the identical experimental setup (hyperparameter values were chosen utilizing the identical strategy as the suggested technique). We employed activations on the deeper layer because it contains more high-level information than the preceding levels (final FC layer). After using activation functions, these layers combine the input features' global spatial positions to yield separate features (i.e., Alexnet and Shufflenet produces a total of 1000 and 544 features). In order to conduct this experiment, we aggregated the images from three publicly accessible datasets and utilized the combined dataset to verify how well our model performed (dataset 1). Training sets make up 80% of the dataset, whereas testing sets make up 20%. Precisely, we used a total of 4000 CRIs from the "COVID-19 Chest Radiography database" dataset (2000 COVID-19, 2000 LO). We applied augmentation on all images of "Chest X-ray (COVID-19 & Pneumonia)" and used 4420 normal and 4456 pneumonia chest radiographs in this experiment. Furthermore, we have used 1220 TB CRIs of the "Lung Disease Dataset (4 types)" dataset. We utilized 11,277 CRIs (1600 COVID-19, 976 TB, 1600 LO, 3536 normal, and 3565 pneumonia chest radiographs) for training our model. The outstanding 2819 CRIs (400 COVID-19, 244 TB, 400 LO, and 884 normal, 891 pneumonia chest radiographs) are utilized for the testing of our model. According to the results (Table 10), as shown in Figure 8, the deep features of these DL-based models and the SVM approach yielded less accurate results (in terms of precision, accuracy, recall, and F-measure) when contrasted to DeepLungNet. The suggested DeepLungNet strategy successfully extracts more distinctive characteristics from the chest radiographs, and, as a consequence, the new approach outperformed the existing methodology in terms of chest radiograph classification to identify TB, LO, pneumonia, normal, and COVID-19-infected people. We achieved the extraction of more strong and more detailed features by using small kernels with 1 × 1 and 3 × 3 dimensions. We used filters of different sizes, i.e., 7 × 7, 3 × 3, and 1 × 1, to extract both global and local features. Additionally, the suggested model's batch normalization approach regularizes and decreases the generalization error and normalizes the inputs to a layer for each mini batch.

**Table 10.** Comparison with CNN + SVM approaches.

| Model | Accuracy | Precision | Recall | F-Measure |
|---|---|---|---|---|
| Alexnet | 97.12 | 92.2 | 91.4 | 91.79 |
| Squeezenet | 97.30 | 92.4 | 92.8 | 92.59 |
| Mobilenetv2 | 97.06 | 92.8 | 92.8 | 92.8 |
| Shufflenet | 97.24 | 93.00 | 93.00 | 93.00 |
| GoogleNet | 97.13 | 93.2 | 92.8 | 92.99 |
| DeepLungNet | 97.47 | 93.2 | 93.6 | 93.4 |

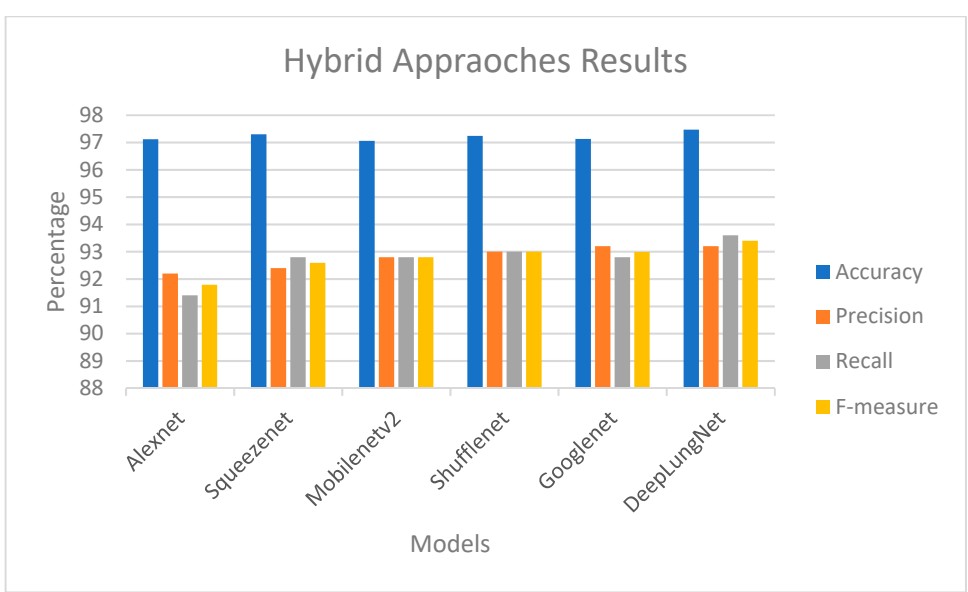

**Figure 8.** Bar chart of comparison with hybrid approaches.

### 4.3.5. Performance Evaluation on "Tumor Classification Data" Dataset

According to experimental studies, the DeepLungNet model works well for a variety of CRIs datasets (lung disease classification); however, additional validation on various datasets is needed to demonstrate our model's flexibility, robustness, and power across all domains. We evaluated the proposed DeepLungNet system utilizing benchmark medical images with the purpose of classifying brain tumors. Brain tumors were divided into benign and malignant categories using this study's "Tumor Classification Data" (dataset for tumor classification), which is freely accessible on the Kaggle website [46]. This collection includes images of the healthy brain together with MRI scans of tumors, both malignant and benign. The dataset is separated into three subfolders, malignant, normal, and benign, in each of the two directories train and test. The only images we used for training and testing our model were the 350 benign and 350 malignant MRI scans from the training folder, i.e., no other images were used. While we only need 70 images of each class for testing, the remaining 280 images of each class were used to train our model. To train our model, we used the exact same experimental setup as that shown in Table 2. For training, our framework required 29 min and 44 s. The outcomes showed that the suggested DeepLungNet technique worked as intended, as evidenced by the achievement of f1-scores of 94.76%, 93.3%, 93.5%, and 93.4% in the relevant medical area. This demonstrates the suggested strategy's efficacy.

### 4.3.6. Lung Disease Classification Comparison with State-Of-The-Art Approaches

This study attempts to validate the DeepLungNet framework's effectiveness in recognizing and categorizing lung illnesses from CRIs. Table 9 displays the findings of our comparison between the suggested strategy and the most recent DL frameworks. The comparative procedure was assessed on the basis of its architectural performance, dataset utilized, DL techniques used, and the number of classes. Using eight pre-trained CNNs, the authors of [47] divided various lung illnesses into pneumonia, TB, pneumothorax, and COVID-19. The categorizing process was split into two steps. During the phase of training, the CNNs were trained using a mini-batch size of 32 and Adam optimizer with a maximum epoch of 30. During the categorizing stage, these trained frameworks were utilized to categorize diseases. In each phase, the dataset was colored-preprocessed, the size of images was reduced, and the data augmentation was applied to increase the number of images. Darknet-19, Alexnet, Darknet-53, GoogleNet, Densenet-201, InceptionResnetV2, Resnet-18, and MobilenetV2 were the eight pre-trained CNNs. One of these frameworks, Densenet-201, has the best accuracy rating at 97.2%. The authors in [48] suggest a DL design for the multi-class categorization of LC, LO, pneumonia, TB, and COVID-19. To

meet the DL requirements, enormous chest x-ray images, including 10,192 normal shots, 20,000 LC images, 1400 TB images, 5870 pneumonia images, 3615 COVID-19 images, and 470 and 6012 LO images, were compressed, standardized, and randomly split. For classification, they employed a pre-trained method comprising VGG19 and three CNN blocks for feature extraction and a FC layer for classification. The experimental results showed that the suggested CNN + VGG19 performed better than other works with a 96.48% accuracy rate. In [49], the authors proposed a straightforward CNN for diagnosing infection on CRIs and evaluated it using 7132 CXR images from publicly available datasets. The results were additionally interpreted and explained to make them more intelligible using Gradient-weighted Class Activation Mapping (Grad-CAM), Shapley Additive Explanation (SHAP), and Local Interpretable Modelagnostic Explanation (LIME). ConV features were first developed to gather thorough object-based data. Then, utilizing shapely values from SHAP, expectedness findings from LIME, and heatmap from Grad-CAM, the black-box technique of the DL framework was investigated; this resulted in an average validation and test accuracy of 94.54% and 94.31%, respectively, for 10-fold cross-validation.

Note that since we do not have the same dataset or size, our comparison may not be fair. Table 11 displays the results of some of the most modern approaches for identifying or categorizing various lung illnesses. However, the anticipated system outperformed the current models with an average accuracy of 97.47%. It is crucial to emphasize that because findings were generated using various datasets, our comparison may not be fair. Furthermore, to the best of our knowledge, no research work in the past has classified lung diseases into LO, pneumonia, VP, BP, TB, and COVID-19. This comparison also demonstrates how effective the DeepLungNet framework is when compared to alternative strategies. It is important to note that these methods use deeper frameworks than the proposed ones, which can unavoidably result in overfitting, making them more computationally expensive. The usefulness of the suggested technique and its extra benefits, including computer efficiency, are demonstrated by these studies. All CNNs layers' biases are not active because the proposed DeepLungNet model only comprises twenty learnable layers, followed by BN and the LReLU layers. We can therefore conclude that the suggested DeepLunNet approach is more successful and efficient at classifying different lung diseases.

**Table 11.** Lung diseases classification performance comparison with state-of-the-art approaches.

| S. No | Work | Classes | Medical Images | Method | Year | Accuracy (%) |
|---|---|---|---|---|---|---|
| 1 | Karaddi et al. [47] | TB, pneumonia, normal, COVID-19, and pneumothorax | 3500 CRIs | Eight pre-trained models including Alexnet, Mobilenetv2, and GoogleNet | 2023 | 97.2 |
| 2 | Alshmrani et al. [48] | LO, normal, pneumonia, COVID-19, TB, and LC | 47,089 CRIs | VGG19 + CNN | 2023 | 96.48 |
| 3 | Bhandari et al. [49] | TB, normal, pneumonia, and COVID-19 | 7132 CRIs | CNN + a black box strategy with XAI. | 2023 | 94.54 |
| 4 | This work | Pneumonia, LO, normal, COVID-19, VP, BP, and TB | Dataset 1 with 14,096 CRIs | DeepLungNet model | 2023 | 97.47 |

## 5. Discussion

The aim and key goal of this paper was to present a DL-based framework for effective lung disease classification and identification including LO, pneumonia, TB, VP, COVID-19, and BP from chest radiographs. Because DL approaches provide better results for the

classification or detection of different diseases of both plants and humans [50–54], we have created an end-to-end solution that does not employ feature extraction or selection. We validated the robustness and generalizability of our suggested technique using two datasets that contained photographs from various databases. In this work, we suggested a DeepLungNet DL-based framework that, when trained on chest radiographs, surpasses competing models in terms of accuracy (97.47). The framework's testing and training accuracy increases and its training and testing loss rapidly decreases after each epoch. The proposed framework is evaluated against both state-of-the-art frameworks that may be presented in the past and hybrid methods (DL + SVM). To evaluate the system's effectiveness and generalizability, we evaluated it using the "Lemon Quality Dataset," a popular and openly available Kaggle dataset from the agriculture domain. The suggested framework outperforms innovative and hybrid techniques and works well.

Since the DeepLungNet framework employed the LReLU activation function instead of the ReLU activation function, our research methodology performs well. We also used the LReLU activation function to address the issue of dying ReLU. In the event of a dying ReLU issue, the DL framework will remain inactive. Using an LReLU, we applied the DeepLungNet approach that was suggested to resolve this issue. When the unit is not active, the LReLU activation mechanism permits for a non-zero (small) gradient. So, it continues to learn rather than coming to a halt or running into a brick wall. As a result, the proposed DeepLungNet model's lung disease classification performance is improved by the LReLU activation function's enhanced feature extraction capability. The vanishing gradient and degradation issues are resolved by DeepLungNet's skip connections method. Each layer that impairs the framework's effectiveness will be skipped, and the gradient will have access to an alternative shortcut path. Learning does not decrease from the first layers to the last layers since the skip connection transfers the output from a previous layer to a following layer. These results are further explained by the fact that our suggested method can effectively extract the most robust, distinctive, and in-depth features to represent the CRI for exact and reliable categorization. Color, edges, and other (low-level) features are extracted by the first convolutional layers. On the other hand, higher layers are in charge of detecting high-level features, such an anomaly in the CRIs. Furthermore, our architecture is based the following concepts. We used filters of different sizes, i.e., $7 \times 7$, $3 \times 3$, and $1 \times 1$ to extract both local and global features. The max-pooling layer in our model aids in the reduction in model dimensions and parameters and the retention of critical feature information. The model also lessens the calculations, i.e., computation cost (to speed up training) by using group ConV operations. A 50% dropout rate is used to reduce co-adaptation and overfitting. When many neurons in a layer extract extremely comparable or same deep features from the input images or data, this is referred to as co-adaptation, which leads to overfitting. Moreover, the BN is utilized to speed up training, standardize the inputs, stabilize the framework, reduce the number of epochs, and provide regularization to prevent the model from overfitting.

It is time consuming and requires a lot of effort to detect lung problems. The images from chest radiographs are also less clear due to noise and fluctuating contrast. As a result, it became difficult for professionals to immediately inspect the CRI. This study provides an automated system for classifying lung disorders that aid in the early detection of lung ailments. This method significantly enhances patient survival and treatment options. The suggested approach provides a trustworthy and efficient way to recognize lung conditions on chest radiographs, supporting the physician in making quick and accurate decisions.

Although the suggested strategy yielded good outcomes, we pointed out a few flaws and made some recommendations for future investigations. The proposed method is unable to categorize many lung disorders, such as pneumothorax, LC, asthma, etc. How successfully the system detects lung disorders when using additional imaging modalities, such as computer tomography, is uncertain from the proposed DeepLungNet technique (CT scans). In the suggested method, image data are repeatedly divided into a test set (20%) and a training set (80%). Yet, different divisions can lead to various consequences. Despite the

fact that our technique accomplished exceptionally well on two CRIs datasets, this study's conclusions have not been validated in real clinical investigations. We will try to employ the suggested approaches to demonstrate the effectiveness of the DeepLungNet algorithm using larger and more varied datasets in the future to resolve the above-mentioned limitations. Just now, we contrasted the effectiveness of our framework and method with hybrid methods, and in the future, we will evaluate the effectiveness of our method with alternative TL-based methods in which we will utilize the FC layer for classification rather than the SVM. Future work will examine how well the suggested model performs in classifying CRIs into more precise categories, such as pneumothorax, LC, asthma, etc., by incorporating data from additional research datasets. In order to use the proposed DeepLungNet model in practical applications to diagnose diseases such as TB, breast cancer, LO, etc., we plan to test its generalizability in the future using more datasets on lung diseases or other datasets (detection and bone crack detection datasets) in the medical domain using CT scans, MRI images, etc. Additionally, in order to validate the outcomes of the suggested method, we wish to judge the DeepLungNet technique using actual clinical cases.

## 6. Conclusions

The early identification and detection of lung disorders is essential to reducing mortality rates and aiding medical personnel. This work uses CRIs to design and test a multiclass categorization of chest disorders based on a DL architecture for TB, LO, VP, BP, pneumonia, normal, and COVID-19. Following image scaling, the generated images were fed into a DeepLungNet framework created to identify various lung illnesses for this purpose. Our DeepLungNet model outperformed other current hybrid techniques with an accuracy of 97.47% for CRIs classification (lungs diseases detection). Our thorough testing has shown that our suggested model performs better than other modern methods. The proposed DeepLungNet identification and classification approach is anticipated to create a framework for classifying lung diseases and to lessen the workload and viral spread associated with COVID-19 medical diagnosis. The proposed DeepLungNet framework can automatically identify lung diseases from CRIs without the need for any physical feature extraction methods because it features an end-to-end learning procedure. A quick and automatic system helps a proficient radiologist in this way by acting as a decision support system. Misdiagnosis can be prevented, and radiologists' efforts can be decreased. Despite the accomplishment of the presented methodology, various DL-based approaches for identifying lung disorders will be created in subsequent works to enhance the DeepLungNet approach's performance.

**Author Contributions:** N.U. and I.A., developed the method; N.U., M.M. and S.A.C., performed the experiments and analysis, and N.U., M.M., I.A. and S.A.C. wrote the paper. All authors have read and agreed to the published version of the manuscript.

**Funding:** This work was supported by Princess Nourah bint Abdulrahman University Researchers Supporting Project number (PNURSP2023R239), Princess Nourah bint Abdulrahman University, Riyadh, Saudi Arabia. The authors extend their appreciation to the Deanship of Scientific Research at King Khalid University for funding this work through large group Research Project under grant number RGP2/249/44.

**Institutional Review Board Statement:** Not applicable.

**Informed Consent Statement:** Not applicable.

**Data Availability Statement:** The datasets used in this investigation are available in the following link; [1] https://www.kaggle.com/datasets/tawsifurrahman/covid19-radiography-database (accessed on 4 September 2022). [2] https://www.kaggle.com/datasets/prashant268/chest-xray-covid19-pneumonia (accessed on 10 November 2022). [3] Datset. Available online: https://www.kaggle.com/datasets/omkarmanohardalvi/lungs-disease-dataset-4-types (accessed on 23 October 2022). Dataset. [4] Available online: https://www.kaggle.com/datasets/yusufemir/lemon-quality-dataset (accessed on 24 November 2022).

**Acknowledgments:** The authors are grateful to the Princess Nourah bint Abdulrahman University Researchers Supporting Project (PNURSP2023R239), Princess Nourah bint Abdulrahman University, Riyadh, Saudi Arabia. The authors extend their appreciation to the Deanship of Scientific Research at King Khalid University for funding this work through large group Research Project under grant number RGP2/249/44.

**Conflicts of Interest:** The authors declare no conflict of interest.

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
