# Peer review of "DeepLungNet: An Effective DL-Based Approach for Lung Disease Classification Using CRIs"

_electronics, doi:10.3390/electronics12081860_

Round 1
Reviewer 1 Report
Please provide theoretical details of why your method is better than others according to the table no 8.
Check the numbering of tables and figures, because figure 2 is missing.
The caption for the diagram of accuracy and loss at page number 15 is missing in section 4.32.
Check the paper for spelling and other grammatical mistakes.
The figures’ resolution can be improved
Heading 4 must be changed to just Results and the manuscript has a separate discussions section (i.e., section 5)
Knowledge gaps and research gaps in this domain can be highlighted.
The results section should contain just results, all explanations should be gathered in the Discussion section (e.g., description of filters and BN, etc). more specifically, this should be a part of the discussion section
“We achieved the extraction of more strong and more detailed features by using small kernels with 1 × 1and 3 × 3 dimensions. Additionally, the suggested model's batch normalization approach regularizes, decreases generalization error, and normalizes the inputs to a layer for each mini-batch.”
Author Response
Thank you for allowing a resubmission of our manuscript, with an opportunity to address the reviewers’ comments and improve the overall quality of the paper by incorporating all the comments from the reviewers. Also, we highlighted the changes in the revised manuscript to make it visible to the reviewers.
We are uploading (a) our point-by-point response to the comments (below) (response to reviewers), (b) an updated manuscript.

Reviewer 2 Report
In this paper, DeepLunNet deep learning (DL) framework is proposed for Lung Disease Classification Using CRIs. This paper is well organized and can be considerd for publication after major modifications listed as follows:
1-There is no figure2 in the manuscript.
2- There are two fig3, which should be modified.
3.Fig.5 in line 542 has no caption number, which should be modified.
4. Citation to websites like references: [2] , [10], [34-36] should be according to the journal format and time should be added.
5. Many parts of paper are copied from [R1], without any changes, which should be modified.
[R1] Ullah N, Khan JA, Almakdi S, Khan MS, Alshehri M, Alboaneen D, Raza A. A novel CovidDetNet deep learning model for effective COVID-19 infection detection using chest radiograph images. Applied Sciences. 2022 Jun 20;12(12):6269.
6. Introductions section should be improved and deep learning approaches should be investigated maybe below paper is helpful: Artificial intelligence and COVID-19: deep learning approaches for diagnosis and treatment. IEEE Access. 2020.
7. Equations which are not belong to authors should be cited and all equations should be written like eq (1).
Author Response

(The authors gave the same response as above.)

Reviewer 3 Report
This paper proposes the DeepLungNet model, consisting of 20 learnable layers, to classify lung diseases and to lessen the workload in medical diagnosis.
First of all, the authors did not mention how they overcome the data imbalance issue. Please specify it.
Second, the References Section is not properly formatted. For example, Items 3 and 10 are not properly cited. Also, the item 1 and item 2 use different styles. Please correct them.
Third, throughout the manuscript, authors used relu, ReLu, and ReLU interchangeably. Please be consistent.
Fourth, the meta data for the image dataset is not provided. Please add this information.
Fifth, the values on page 2 are misleading. For example 60, 719, 433 should be 60,719,223. Please update them.
Sixth, the subscripts and superscripts of Equation (1) should be corrected in lines 343, and 344.
Besides, there are some typos points to be corrected:
Page 3, line 140: “DeepLungNETDL” should be “DeepLungNET DL.”
Page 4, line 161: “DeepLungNetmodel” should be “DeepLungNet model.”
Page 7, line 310: “20 and 20” should be “-20 and 20.”
Page 8, line 332: “ConVand” should be “Conv and.”
Author Response

(The authors gave the same response as above.)

Reviewer 4 Report
Below are the comments for DeepLungNet: An Effective DL-based Approach for Lung Disease Classification Using CRIs
- It seems confusing at time about how many classes are being considered. Abstract has 6 classes but Table 5 lists only 5 classes. Please be consistent. Also, it may be my personal preference but having consistent order of the classes will help the readers.
- Replace words pictures, photos, pics by images in the paper.
- Line 298: Lists 80 % data for training and 20 % for testing. This is reversed for line 324.
Line 343: use the same format for terms used in equation (1) to describe them. In terms of subscript and superscripts.
For section 4. Please list the Database 1 and 2 information in the tabular format. The table should include total images, training and testing for each class after data augmentation.
- Plot accuracy, prevision, recall and F1 scores graphically for Tables 7 and 8.
- Table 9 column heading Date should be Year
- It is not clear why is Lemon dataset used to check generalizability. Is it possible to try something from medical field such as brain tumor or bone crack detection ?
Thank you!
Author Response

(The authors gave the same response as above.)

Reviewer 5 Report
The paper proposes a DeepLungNet for COVID-19 detection on chest radiograph images. The paper is well structured and presented in overall. Some suggestions for improvements are:
1. Please remove inappropriate self citation: [37] and [43]
2. Proofreading is required as there are some spelling errors.
3. It would be good to present the hyperparameter tuning results of some important hyperparameters.
Thank you.
Author Response

(The authors gave the same response as above.)

Round 2
Reviewer 1 Report
My recommendation is "Accept in present form".
Reviewer 2 Report
The authors have answered most of my concerns.
Reviewer 3 Report
The manuscript has been significantly improved and can be accepted in present form.
Reviewer 4 Report
Thank you for considering the comments and revising the manuscript.